# Variable Selection and Regularization in Quantile Regression via Minimum Covariance Determinant Based Weights

**DOI:** 10.3390/e23010033

**Published:** 2020-12-29

**Authors:** Edmore Ranganai, Innocent Mudhombo

**Affiliations:** 1Department of Statistics, University of South Africa, Florida Campus, Private Bag X6, Florida Park, Roodepoort 1710, South Africa; 2Department of Accountancy, Vaal University of Technology, Vanderbijlpark Campus, Vanderbijlpark 1900, South Africa; innocentm@vutcloud.onmicrosoft.com

**Keywords:** weighted quantile regression, RIDGE penalty, LASSO penalty, elastic net penalty, high leverage points, collinearity influential points, minimum covariance determinant

## Abstract

The importance of variable selection and regularization procedures in multiple regression analysis cannot be overemphasized. These procedures are adversely affected by predictor space data aberrations as well as outliers in the response space. To counter the latter, robust statistical procedures such as quantile regression which generalizes the well-known least absolute deviation procedure to all quantile levels have been proposed in the literature. Quantile regression is robust to response variable outliers but very susceptible to outliers in the predictor space (high leverage points) which may alter the eigen-structure of the predictor matrix. High leverage points that alter the eigen-structure of the predictor matrix by creating or hiding collinearity are referred to as collinearity influential points. In this paper, we suggest generalizing the penalized weighted least absolute deviation to all quantile levels, i.e., to penalized weighted quantile regression using the RIDGE, LASSO, and elastic net penalties as a remedy against collinearity influential points and high leverage points in general. To maintain robustness, we make use of very robust weights based on the computationally intensive high breakdown minimum covariance determinant. Simulations and applications to well-known data sets from the literature show an improvement in variable selection and regularization due to the robust weighting formulation.

## 1. Introduction

Variable selection and robust estimation procedures are an important consideration of multiple regression analysis in the presence of predictor space data aberrations (high leverage points (outliers in the *X*-space) and multicollinearity) as well as response variable (*Y*-space) outliers. It is well known that the least squares (LS) are susceptible to both data aberrations in the predictor space and response variable (*Y*-space) outliers. To counter the influence of *Y*-space outliers, alternative robust procedures have been developed in the literature. One such attractive robust procedure is quantile regression (QR) [1]. In addition to being robust, QR is more versatile than the LS. This is due to the fact that, while the LS procedure models the conditional mean (E(Y|X)) (center of distribution), the QR procedure is able to detect heterogeneous effects of predictors at different quantile levels of the outcome as it models the conditional quantiles ($Q_{Y|X}\left(\tau \right)$ for 0<τ<1) of the response variable *Y* given the predictors *X* over the entire range of quantiles in (0,1) [2]. Regression quantiles (RQs) are optimal solutions to an optimization problem obtained using linear programming (LP) algorithms [3]. An RQ at the τ=0.5 quantile level corresponds to the well-known least absolute deviation (LAD) estimator, i.e., the $\ell_{1}$ estimator. Although RQs are robust to *Y*-space outliers, on the other hand, they are susceptible to high leverage points as their influence function are bounded in the *Y*-space but unbounded in the *X*-space. Amongst the numerous sources of multicollinearity, some high-leverage observations tend to influence the eigen-structure of the predictor matrix thereby creating multicollinearity or hiding it [4]. Such high leverage points are referred to as collinearity-influential points. However, not all high leverage observations are collinearity influential points. As remedies to high leverage points influences, weighted LAD (WLAD) [5] and weighted QR (WQR) have been suggested in the literature [6].

In both variable selection and regularization, in order to enhance the prediction accuracy and interpretability of statistical models, the RIDGE type [7], the least absolute shrinkage and selection operator (LASSO) type [8,9,10] penalties and a hybrid of these two penalties, viz., elastic net (E-NET) penalty [11,12] as well as the smoothly clipped absolute deviation (SCAD) penalty [13,14] have been suggested in the literature. Notable extensions of the LASSO penalty in the literature are the adaptive LASSO proposed by [15], fused LASSO [16], and group LASSO [17]. In high-dimensional sparse models where ordinary quantile regression is not consistent, [18] proposed the $\ell_{1}$-penalized QR. The author of [14] developed, upon the procedure of [19] on model selection in composite quantile regression (CQR) and suggested weighted CQR (WCQR), a procedure based on data driven efficient weights, as the former researcher’s equal weight property procedure lacked optimality. To mitigate against the undesirable effects of high leverage points in variable selection in the $\ell_{1}$ estimator ($Q_{Y|X}\left(0.5\right)$), the weighted LAD-LASSO (WLAD-LASSO) procedure has been suggested [20,21]. Few variable selection procedures based on WQR have been suggested in the QR regression framework in different settings. We generalize the WLAD-LASSO approach to penalized WQR to mitigate against collinearity influential points and high leverage points in general, and maintain robustness via use of very robust weights.

In summary, the motivations of this study are premised on the following:The generalization of WLAD-LASSO [20,21] (in addition, we also include the RIDGE and the E-NET penalties) procedure to the QR framework, i.e., to penalized WQR as each RQ (including the LAD estimator) is a local measure, unlike the LS estimator, which is a global one.Rather than carrying an "omnibus" study of penalized WQR as in [20,21], we carry out a detailed study by distinguishing different types of high leverage points viz.,
–Collinearity influential points which comprise collinearity inducing and collinearity hiding points.–High leverage points which are not collinearity influential.Taking advantage of high computing power, we make use of the very robust weights based on the computationally intensive high breakdown minimum covariance determinant (MCD) method rather than the well-known classical Mahalanobis distance or any LS based weights as in [20] which are amenable to outliers.

The remainder of this article is structured as follows. Some preliminaries on QR and variable selection in QR are discussed in Section 2. Variable selection in WQR as well as motivation for our choice of weights are detailed in Section 3 while simulation studies are detailed in Section 4. In Section 5, applications to two well-known data sets from the literature are detailed. Lastly, Section 6 concludes the paper.

## 2. Preliminaries

### 2.1. Quantile Regression

Consider the linear regression model given by
(1)$$y_{i}=x_{i}^{\prime }\beta +\epsilon_{i},\;\;for\;\;i=1,2,3,...,n$$
where $y_{i}$ denotes the value of the response variable vector Y for the ith observation, $x_{i}={\left(x_{i1},x_{i2},\ldots ,x_{ip}\right)}^{\prime }$ denotes the vector of *p* predictor variables from the n×p design matrix X excluding the intercept, $\beta ={\left(\beta_{1},\ldots ,\beta_{p}\right)}^{\prime }$ denotes a p×1 vector of yet to be estimated unknown regression coefficients (parameters) and $\epsilon_{i}$ denotes the value of the ith random error term, with cumulative distribution function *F* ($\epsilon_{i}∼F$).

QR is based on an optimization problem which can be solved by linear programming techniques, viz.,
(2)$$\hat{\beta }\left(\tau \right)=\underset{\beta \in R^{p}}{\mathrm{argmin}}\Sigma_{i=1}^{n}\rho_{\tau }\left(y_{i}-x_{i}^{\prime }\beta \left(\tau \right)\right),\;\;i=1,2,\ldots ,n$$
where $\rho_{\tau }\left(u\right)=u\left[\tau -I\left(u<0\right)\right]\equiv u\left[\tau .I\left(u\geq 0\right)+\left(\tau -1\right).I\left(u<0\right)\right]$ and $\hat{\beta }\left(\tau \right)$ denotes the $\tau^{th}$RQ.

### 2.2. Variable Selection in Quantile Regression

In this section, we discuss QR variable selection. We specifically present variable selection using LS-RIDGE [7], LASSO [9], adaptive LASSO [15], and E-NET [11] in a QR scenario. Firstly, we consider QR penalized with the RIDGE penalty [7] denoted by QR-RIDGE. The QR-RIDGE is given by the minimization problem
(3)$$\hat{\beta }\left(\tau \right)=\underset{\beta \in R^{p}}{\mathrm{argmin}}\Sigma_{i=1}^{n}\rho_{\tau }\left(y_{i}-x_{i}^{\prime }\beta \left(\tau \right)\right)+\lambda \Sigma_{j=1}^{p}\beta_{j}^{2},j=1,2,\ldots ,p,\;i=1,2,\ldots ,n$$
where λ is a positive ridge parameter in the range 0<λ<1 and other terms are as defined in Equation (Equation 3). Many variations of λ have been used in literature (see [7,22,23,24,25,26]). QR with the RIDGE ($\ell_{2}$-squared) penalty has been proposed as a remedy to the multicollinearity problem [25]. The presence of multicollinearity results in undue large sample variance resulting in unreliable inference and prediction.

Secondly, we consider QR variable selection procedure which uses the LASSO ($\ell_{1}$) penalty [9] denoted by QR-LASSO. The QR-LASSO is then given by the minimization problem
(4)$$\hat{\beta }\left(\tau \right)=\underset{\beta \in R^{p}}{\mathrm{argmin}}\Sigma_{i=1}^{n}\rho_{\tau }\left(y_{i}-x_{i}^{\prime }\beta \left(\tau \right)\right)+n\lambda \Sigma_{j=1}^{p}\left|\beta_{j}\right|,j=1,2,\ldots ,p,\;i=1,2,\ldots ,n$$
where λ is the tuning parameter that shrinks some beta coefficients towards zero, the second term is the penalty term, and other terms are as defined in Equation (Equation 3). This $\ell_{1}$-penalized QR may be superior to the $\ell_{2}$-squared penalized QR in Equation (Equation 3) in some instances.

Considering the more recent adaptive LASSO penalty by [15] in a penalized QR scenario, the tuning parameter is no longer constant (λ) but $\lambda_{j}$ for j=1,2,…,p. The minimization problem becomes
(5)$$x\hat{\beta }\left(\tau \right)=\underset{\beta \in R^{p}}{\mathrm{argmin}}\Sigma_{i=1}^{n}\rho_{\tau }\left(y_{i}-x_{i}^{\prime }\beta \left(\tau \right)\right)+n\Sigma_{j=1}^{p}\lambda_{j}\left|\beta_{j}\right|,j=1,2,\ldots ,p,\;i=1,2,\ldots ,n$$
where $\lambda_{j}$ is the jth tuning parameter that shrinks some predictor variables to zero and other unknowns are as defined in Equation (Equation 4).

We also present a penalized QR procedure that uses the elastic NET penalty from [11] (E-NET-penalized QR which is best suitable for applications with unidentified groups of predictors). The E-NET penalized QR is given by
(6)$$\hat{\beta }\left(\tau \right)=\underset{\beta \in R^{p}}{\mathrm{argmin}}\Sigma_{i=1}^{n}\rho_{\tau }|y_{i}-x_{i}^{\prime }\beta \left(\tau \right)|+\alpha \lambda \Sigma_{j=1}^{p}\left|\beta_{j}\right|+\left(1-\alpha \right)\lambda \Sigma_{j=1}^{p}\beta_{j}^{2},j=1,2,\ldots ,p,\;i=1,2,\ldots ,n$$
where α∈[0,1] and λ is the tuning parameter for the second and third terms, respectively. Note that, for α=0, the E-NET penalty reduces to the RIDGE penalty while, for α=1, it reduces to the LASSO penalty. In some instances, the E-NET based procedure performs better than its ridge and LASSO counterparts [11]. Since QR is prone to outliers in the predictor space, weighted QR has been proposed as a remedy to high leverage points [6]. In the subsequent section, we motivate the choice of weights used in robust variable selection in QR framework.

## 3. Variable Selection and Regularization in Weighted Quantile Regression

### 3.1. Choice of Weights for Downweighing High Leverage Observations Motivation

In the LS case, statistics of the hat (projection) matrix $h_{i}=x_{i}{\left(X^{\prime }X\right)}^{-1}x_{i}^{\prime }$ [4] have been used as standard tools to generate weights for weighted LS (WLS) estimation. Although this approach is both mathematically and practically tractable, such estimators have a breakdown point of only 1/n so a single high leverage point can completely dominate the ensuing estimates. Furthermore, such weights may suffer from the masking and swamping effect associated with the LS. Permitting contamination in both the predictor and response variables results in the breakdown point of LAD (and hence QR) estimator being the same as that of LS, 1/n (see, e.g., [27]). To circumvent the undesirable effects of both outliers and high leverage points, Ref. [5] proposed a weighted version of an LAD (WLAD) estimator. The weights
(7)$$\omega_{j}=min\left(1,\frac{p}{RD{\left(x_{j}\right)}^{2}}\right),\;\;j=1,2,\ldots ,n$$
of this estimator are based on the computationally intensive high breakdown Minimum Covariance Determinant (MCD) method [28]. $RD\left(x_{j}\right)=\sqrt{{\left(x_{j}-\hat{\mu }\right)}^{\prime }{\hat{\Sigma }}^{-1}\left(x_{j}-\hat{\mu }\right)}$ is the robust distance (a modification of the classical Mahalanobis distance), $\hat{\mu }$ is the center of the smallest ellipsoid (whose classical covariance matrix has the lowest possible determinant) containing half (or *h* observations as defined by the user) of the observations of the design matrix X, and $\hat{\Sigma }$ is their covariance matrix multiplied by a consistency factor [29]. To side step the huge computational load associated with this weight, Ref. [30] suggested $\omega_{i}=\sqrt{min_{j}\left(h_{j}/h_{c_{i}}\right)}$, where $h_{c_{i}}=x_{i}{\left(X_{c}^{\prime }X_{c}\right)}^{-1}x_{i}^{\prime }$ is the $i^{th}$ leverage point relative to the clean subset $X_{c}$ (without high leverage points). However, in this study, we make use of $RD(x_{j})$ due to the existing computer power (efficient algorithms) as well as its robustness to generalize the WLAD concept to the whole set of weighted conditional quantiles, i.e., weighted regression quantiles (WRQs).

Generalizing the [20] WLAD regression estimator and making use of the MCD based weights, we suggest a WQR estimator as a minimization problem given by
(8)$$\hat{\tilde{\beta }}\left(\tau \right)=\underset{\beta \in R^{p}}{\mathrm{argmin}}\Sigma_{i=1}^{n}\omega_{i}\rho_{\tau }\left|y_{i}-x_{i}^{\prime }\beta \left(\tau \right)\right|\;\;i=1,2,\ldots ,n,$$
where $\omega_{i}$ and $\rho_{\tau }\left(.\right)$ are as defined in Equations (2) and (7), respectively.

### 3.2. Penalized Weighted Quantile Regression

Building on Equation (Equation 8), we suggest a penalized weighted QR (WQR) variable selection procedure using MCD based weight 7 to counter the undesirable influences of high leverage observations. We achieve robustness of WQR in the *X*-space due to the robustness of the this MCD based weights chosen appropriately as in the LS case (see also [5,30]). The then suggested penalized WQR variable selection procedures are based on three penalty functions, viz., the RIDGE, LASSO, and the E-NET penalties bringing about the WQR-RIDGE, WQR-LASSO, and WQR-E-NET estimators, respectively.

Let $\omega_{i}$ be a robust weight. We use $\omega_{i}$ in our proposed quantile variable selection procedures. The incorporation of WLAD-LASSO in the procedure inherits *X*-space robustness property as in [20]. The weight $\omega_{i}$ was discussed in Section 3. First, we propose the WQR-RIDGE given by
(9)$$\hat{\tilde{\beta }}=\underset{\tilde{\beta }\in R^{p}}{\mathrm{argmin}}\Sigma_{i=1}^{n}\omega_{i}\rho_{\tau }\left(y_{i}-x_{i}^{\prime }\beta \left(\tau \right)\right)+\lambda \Sigma_{j=1}^{p}\beta_{j}^{2},j=1,2,\ldots ,p,\;i=1,2,\ldots ,n$$
where the terms are already defined. The tuning parameter λ shrinks the coefficients of the predictor variables towards zero.

We take advantage of the properties of WLAD-LASSO [20] and propose a weighted quantile variable selection procedure called WQR-LASSO. The WQR-LASSO procedure is expected to be robust and superior to its penalized QR counterpart, QR-LASSO. The WQR-LASSO is the solution of a minimization problem given by
(10)$$\hat{\tilde{\beta }}=\underset{\tilde{\beta }\in R^{p}}{\mathrm{argmin}}\Sigma_{i=1}^{n}\omega_{i}\rho_{\tau }\left(y_{i}-x_{i}^{\prime }\beta \left(\tau \right)\right)+n\lambda \Sigma_{j=1}^{p}\left|\beta_{j}\right|,j=1,2,\ldots ,p,\;i=1,2,\ldots ,n$$
where tuning parameter λ is constant. In the literature, this procedure is found to perform better than the WQR-RIDGE procedure under deviations from the Normality assumptions.

Lastly, we apply the E-NET penalty to WQR variable selection to bring about the WQR-E-NET procedure. This weighted penalized QR procedure has both the LASSO and RIDGE penalties properties inherent in it [11]. The WQR-E-NET is the solution to a minimization problem given by
(11)$$\hat{\tilde{\beta }}=\underset{\tilde{\beta }\in R^{p}}{\mathrm{argmin}}\Sigma_{i=1}^{n}\omega_{i}\rho_{\tau }\left(y_{i}-x_{i}^{\prime }\beta \left(\tau \right)\right)+\alpha \Sigma_{j=1}^{p}\left|\beta_{j}\right|+\left(1-\alpha \right)\lambda \Sigma_{j=1}^{p}\beta_{j}^{2},j=1,2,\ldots ,p,\;i=1,2,\ldots ,n$$
where α and λ are as in Equation (Equation 6).

### 3.3. Asymptotic Properties

We conveniently decompose the regression coefficient as $\beta ={\left(\beta_{1}^{\prime },\beta_{2}^{\prime }\right)}^{\prime }$, where $\beta_{1}^{\prime }={\left(\beta_{1},\ldots ,\beta_{p_{0}}\right)}^{\prime }$ and $\beta_{2}^{\prime }={\left(\beta_{p_{0}+1},\ldots ,\beta_{p}\right)}^{\prime }$; $x_{i}^{\prime }={\left(x_{i1}^{\prime },x_{i2}^{\prime }\right)}^{\prime }$, where $x_{i1}^{\prime }={\left(x_{i1},\ldots ,x_{ip_{0}}\right)}^{\prime }$ and $x_{i2}^{\prime }={\left(x_{i(p_{0}+1)},\ldots ,x_{ip}\right)}^{\prime }$ such that
(12)$$y_{i}=x_{i}^{\prime }\beta +\epsilon_{i}\equiv x_{i1}^{\prime }\beta_{1}+x_{i2}^{\prime }\beta_{2}+\epsilon_{i},\;\;for\;\;i=1,2,3,\ldots ,n,$$
with true regression coefficient $\beta_{1}$ corresponding to non zero coefficients and $\beta_{2}=0$.

To establish asymptotic normality, suppose for a suitable choice of $\lambda_{n}$ that we now assume the two theoretical conditions to be true as stated in [13], as follows:(i)The regression errors $\epsilon_{i}$’s are i.i.d., with τth quantile zero and continuous, positive density f(.) in a neighborhood of zero (see [31]).(ii)Let $W=diag(\omega_{1},\omega_{2},\ldots ,\omega_{n})$, where $\omega_{i}$ for i=1,2,…,n are known positive values that satisfy $max\left\{\omega_{i}\right\}=O\left(1\right)$.(iii)There exists a positive definite matrix Σ: $lim_{n\to \infty }\frac{\Sigma_{i=1}^{n}\omega_{i}x_{i}^{\prime }x_{i}}{n}=\Sigma$, where $\Sigma_{11}$ and $\Sigma_{22}$ denote the $p_{0}\times p_{0}$ and $\left(p-p_{0}\right)\times \left(p-p_{0}\right)$ top-left and right-bottom submatrices of Σ, respectively.

We first give the following Theorem 1 (oracle property) for the i.i.d. error terms case ($W=I_{n}$) before we consider the non i.i.d. error terms case which concerns this study.

**Theorem** **1.**
*Consider a sample $\left\{\left(x_{i},y_{i}\right),\;\;i=1,\ldots ,n\right\}$ from model (12) satisfying conditions (i) and (iii) (with $W=I_{n}$). If $\sqrt{n}\lambda_{n}\to 0$ and $n^{(\gamma +1)/2}\lambda_{n}\to \infty$, then we have*

*(1) Sparsity: ${\hat{\beta }}_{2}=0$;*

*(2) Asymptotic normality: $\sqrt{n}\left({\hat{\beta }}_{1}-\beta_{1}\right)\overset{d}{⟶}N\left(0,\frac{\tau \left(1-\tau \right)\Sigma_{11}^{-1}}{f{\left(0\right)}^{2}}\right)$.*


In order to extend the conclusions of the i.i.d. error terms case to the non i.i.d. error terms case, we invoke the following assumptions by [32]:(K1)As n→∞$max_{i=1,\ldots ,n}\left\{x_{i}^{\prime }x_{i}/n\right\}\to 0$.(K2)The random error terms $\epsilon_{i}$’s are independent with $F_{i}\left(t\right)=P\left(\epsilon_{i}\leq t\right)$ the distribution function of $\epsilon_{i}$. We assume $F_{i}\left(.\right)$ is locally linear near zero (with a positive slope) and $F_{i}\left(0\right)=\tau$.(K3)Assume that, for each u, $\left(1/n\right)\sum_{i=1}^{n}\psi_{ni}\left(u^{\prime },x_{i}\right)\to \zeta$, where ζ(.) is a strictly convex function taking values in [0,∞) with $\psi_{ni}=\int_{0}^{t}\sqrt{n}\left(F_{i}\left(s/\sqrt{n}\right)-F_{i}\left(0\right)\right)ds$ denoting a convex function for each *n* and *i*.

**Corollary** **1.**
*Under assumptions (ii), (iii), and (K1), Theorem 1 holds provided the non i.i.d. error terms satisfy (K2) and (K3).*


**Remark** **1.**
*The proofs of Theorem 1 and Corollary 1 are outlined in [13] (online supplement materials).*


## 4. Simulation Study

In this section, we perform a simulation study to investigate the finite-sample performance of penalized WQR under the RIDGE, the LASSO, and the E-NET penalty (for α=0.5) functions in terms of the variable selection and regularization of the regression parameters making use of Equations (9)–(11) and the robust MCD based weight $\left(\omega_{j}\right)$ given in Equation (Equation 7) in comparison to their unweighted versions. For brevity, we consider τ=0.5 (the LAD estimator) and τ=0.25 RQ levels. We summarize the simulation results in terms of the percentage of correctly estimated regression models, the average correct zero coefficients ($\beta_{3}$, $\beta_{4}$, $\beta_{6}$, $\beta_{7}$, and $\beta_{8}$) and the average incorrect zero coefficients along with the median of the test error and its respective measure of dispersion,
$$MAD=1.4826{median}_{i=1,\ldots ,n}\left|\epsilon_{i}-{median}_{k=1,\ldots ,n}\left(\epsilon_{k}\right)\right|,$$
where constant 1.4826 is a correction factor which makes the MAD consistent at Normal distributions (see e.g., [29]). The simulation study is designed according to the following scenarios with the predictor matrices of size n×p, p=8 and n=50,100 (but we only give results for n=50 for brevity);

D1− This predictor matrix is obtained from orthogonalization such that $X^{\prime }X=nI$. Using singular value decomposition (SVD), we solve $W=UDV^{\prime }$, where $w_{ij}∼N\left(0,1\right)$ for i=1,…,n and j=1,…,p; **U** and **V** are orthogonal with the diagonal entries of **D** giving the singular (eigen) values of **W**. Then, $X=\sqrt{n}U$ is such that $X^{\prime }X=nI$ due to orthogonality of **U**.D2−has a collinearity inducing point which is D1, but with observation having the largest Euclidean distance from the center of the design space moved 10 units in the *X*-space.D3−has collinearity hiding point which is D1, but with observations having the largest (second largest) Euclidean distance from the center of the design space moved 10 units in the *X*-space.D4−has a collinearity inducing point which is D1, but with observation having the largest Euclidean distance from the center of the design space moved 100 units in the *X*-space.D5−has collinearity hiding point which is D1, but with observations having the largest (second largest) Euclidean distance from the center of the design space moved 100 units in the *X*-space.D6−has (m=5) (n−m)×p correlated $X_{1}$ and m×p leverage contaminated $X_{2}$ sub matrices of D6, i.e., $X=\left(\begin{array}{c} X_{1}\\ X_{2} \end{array}\right)$, where $X_{1}∼N\left(\mu_{1},V\right)$ with $\mu_{1}={\left(0,0,0,0,0,0,0,0\right)}^{\prime }$ and $v_{ij}=0.5^{\left|j-i\right|}$ (0.5 controls the degree of correlation), i,j=1,2,3,4,5,6,7,8, is the ${\left(ij\right)}^{th}$ entry of the covariance matrix V), is the (n−m)×p correlated sub matrix of D6; $X_{2}∼N\left(\mu_{2},I\right)$ with $\mu_{2}={\left(1,1,1,1,1,1,1,1\right)}^{\prime }$ is the m×p leverage contaminated sub matrix of D6.

The predictor matrices D1–D5 are constructed as in [33], while D6 is constructed as in [20].

The regression coefficients with zero intercept, i.e., $\beta_{0}=0$ are

$\beta_{1}={\left(3.5,2,0,0,2.5,0,0,0\right)}^{\prime },\;\beta_{2}={\left(2,1,0,3,1.5,0,1,0\right)}^{\prime }$.

We consider the following error term distribution scenarios;

$\epsilon ∼N(\mu ,\sigma^{2})$, with (μ,σ) choices (0,1) and (0,3).$\epsilon ∼t_{d}$ with choices d=1,6.

The design matrix D1 is used as a baseline, and a schematic representation of D2–D5 departures from it is shown in Figure 1; D2 and D4 have a high leverage point that induces collinearity while D3 and D5 each have a pair of high leverage points that hide collinearity, viz., one for inducing the collinearity and the other for hiding it. On the other hand, D6 has both multicollinearity due the covariance structure, V of the sub matrix $X_{1}$, and high leverage points due to the mean shift in $X_{2}$.

D1–D6 (see, e.g., the response variable $Y={\left(Y_{1},Y_{2}\right)}^{\prime }$ is generated as in [20], i.e., $Y_{1}=X_{1}^{\prime }\beta_{1}+\sigma \epsilon ,\;\epsilon ∼N\left(0,\sigma^{2}\right);\;\sigma =1,\;3$ and $Y_{2}=X_{2}^{\prime }\beta_{2}$;

$Y_{1}=X_{1}^{\prime }\beta_{1}+\sigma \epsilon ,\;\epsilon ∼t_{d};\;\sigma =0.5,\;1$ and $Y_{2}=X_{2}^{\prime }\beta_{2}$.

The number of simulation runs is 200 while the 100-fold cross-validations are employed to obtain the tuning parameters λ. Instead of using cross-validation error metrics independent tuning data sets and testing data sets of size *n* and 100n were generated in the exact way the training data sets were generated (see, e.g., [13]). This simulation study explores these different scenarios in both QR and WQR variable selection procedures using the R-add-on package hqreg. In this package, the regularization parameter lambda is fit using a semismooth Newton coordinate descent algorithm. See [34] for details.

A schematic representation of collinearity influential points is given in Figure 1 below showing the scatter plots representations of predictor matrices D2–D5. The extreme observation in D2 and D4 creates collinearity while the second extreme observation D3 and D5 obscures it. We only consider the Normal distribution at the well-behaved orthogonal predictor matrix D1, and from thereon we only consider the *t* distribution as QR being designed to handle distributions heavier than the Normal one which is handled best by the LS.

### 4.1. Results

D1 SCENARIO

As a point of departure, we consider the well-behaved predictor matrix D1 under the Normal distribution as contrasted with the D1 under the *t* distribution on 1 d.f. (implying outliers). The results given in Table 1 are as expected. At D1 under the Normal distribution, variable (model) selection performs best under the LASSO penalty followed by the E-NET penalty across all models with no marked differences in the median and MAD test error measures indicating the robustness of the QR-LASSO procedure under the *t* distribution. This is further illustrated graphically in Figure 2 and Figure 3 where the QR-LASSO procedure correctly shrinks to zero the zero coefficients {$\beta_{3}$, $\beta_{4}$, $\beta_{6}$, $\beta_{7}$, $\beta_{8}$ } more often than not.

**Remark** **2.**
*The five zero coefficients correspond to the set {$\beta_{3}$, $\beta_{4}$, $\beta_{6}$, $\beta_{7}$, $\beta_{8}$ }, hence the maximum average of correctly/incorrectly selected (shrunk) coefficients is 5 while the set of correctly selected models is given as a proportion, i.e., a %.*


D2 AND D4 SCENARIOS

We consider the introduction of collinearity inducing points in both D2 and D4. The RIDGE penalty performs the worst in every scenario in both penalized QR and penalized WQR in model/variable selection. The model/variable selection pattern at D2 and D4 under both the $t_{6}$ and $t_{1}$ distributions is shown in Figure 4. The dominance of WQR-LASSO followed by WQR-E-NET is clearly depicted. However, the performance of WQR-E-NET is generally better under the $t_{1}$ distribution than it is under the $t_{6}$ distribution. In Figure 4 (lower panels), both the median absolute test error and its MAD measure (in the line graph) show that generally the unweighted versions outperform the weighted ones.

D3 AND D5 SCENARIO

We consider the introduction of collinearity hiding points in both D3 and D5. The performances under the $t_{6}$ and $t_{1}$ distributions are shown graphically in Figure 5. In model/variable selection, throughout all the scenarios, the weighted penalized versions outperform the penalized unweighted versions under the LASSO and E-NET penalties. Amongst the penalized weighted versions, WQR-RIDGE performs the worst while WQR-LASSO performs better than WQR-E-NET. The prediction pattern under the *t* distributions exhibited under D3 and D5 in Figure 5 (lower panel) are different to that exhibited at D2 and D4 in that the MAD error measure is more erratic (but the absolute median error is less erratic) at D3 and D5 but the absolute. In fact, the prediction picture of QR-LASSO and WQR-LASSO at τ=0.5 based on the absolute median error are similar at D3 and D5, whereas, at D2 and D4, WQR-LASSO performs better with respect to this measure.

D6 SCENARIO

The D6-scenario model/variable selection performance outcomes under the *t* distribution is given in Figure 6. In Figure 6, the dominance of the LASSO penalty in both QR and weighted QR is clearly depicted with WQR-LASSO far outperforming QR-LASSO in model/variable selection. Overall, on average, the zero βs are most incorrectly selected under the $t_{1}$ distribution and at σ=1. The prediction pattern under the *t* distribution exhibited under D6 in Figure 6 (lower panel) is slightly poorer under the $t_{1}$ distribution compared to that exhibited under the $t_{6}$ distribution.

## 5. Examples

In this section, we consider two data sets often used to illustrate the efficacy of robust methodologies in mitigating against high leverage points in general as well as collinearity influential points in particular, viz. the [35] and the [36] data sets. In both data sets, the response is generated based on the $t_{1}$ error term distribution in line with testing the efficacy of robust procedures like QR as in [20].

**Remark** **3.**
*The LS procedure is adversely affected by both high leverage points and outliers in the Y-space, hence it consistently gives the worst performance as expected. On the other hand, QR is not affected by the latter data aberrations. Consequently, we mainly focus on the efficacy of penalized WQR at quantile levels τ=0.5 and τ=0.25 in addressing the problem of high leverage points with the $intercept=F^{-1}\left(\tau \right)+\beta_{0}$ corresponding to 0 and −1 under the $t_{1}$ error term distribution, respectively.*


### 5.1. Hawkins, Bradu, and Kass Data Set

We firstly consider the [35] which consists of 75 observations with three predictor variables. The first 14 observations of the 75 observations are high leverage points with the first 13 observations inducing the collinearity while the 14th observation greatly affects the collinearity structure on its own although it is also a collinearity inducing point (see [37]). Figure 7 shows the leverage structure for the predictor variables of this data set based on the robust MCD based distance as well as the classical one. We give results for the full data set and reduced data sets without observations 1–14.

The response variable is generated as $Y_{2}=X_{2}^{\prime }\beta_{2}$ for the first 10 observations and $Y_{1}=X_{1}^{\prime }\beta_{1}+\epsilon ,\;\epsilon ∼t_{1}$ for the remainder of the data, where $\beta_{2}={\left(2,2,0\right)}^{\prime }$ and $\beta_{1}={\left(1,1,0\right)}^{\prime }$ such that $Y={\left(Y_{1},Y_{2}\right)}^{\prime }$

The results for the full data set are given in Table 2.

Similar poor performances of penalized QR at both τ-levels are exhibited across all penalty functions. However, penalized WQR exhibits a drastic improvement at τ=0.5 where penalized WQR is equivalent to unpenalized WQR under both the RIDGE and the E-NET penalties as the λ=0. At τ=0.25, where λ≠0, WQR tends to be too “greedy” under LASSO and E-NET penalties where all the parameters are shrunk to zero.

Results for the reduced [35] data set with a “clean” predictor matrix (without observations 1–14) are given in Table 3. As expected, both QR and WQR select unpenalized models (models with tuning parameter λ=0) except WQR at τ=0.25. QR performs considerably well at both τ levels and across penalty functions. However, there is a marginal improvement in using penalized WQR at τ=0.5 while, at τ=0.25, the LASSO and E-NET penalties are too “greedy”.

### 5.2. Hocking and Pendleton Data Set

While the [35] set is an example of high leverage points that are collinearity inducing points, the [36] data set is an example of high leverage points that are collinearity hiding points. This data set has 26 observations with three predictor variables, $X_{1}$, $X_{2}$, and $X_{3}$, whereby $X_{3}$ is created as a linear combination of $X_{1}$ and $X_{2}$. The response variable is generated as $Y_{1}=X_{1}^{\prime }\beta_{1}+\epsilon ,\;\epsilon ∼t_{1}$ for the first 22 observations and $Y_{2}=X_{2}^{\prime }\beta_{2}$ for the remainder of the data, where $\beta_{1}={\left(3,-2,0\right)}^{\prime }$ and $\beta_{2}={\left(1,1,0\right)}^{\prime }$ such that $Y={\left(Y_{1},Y_{2}\right)}^{\prime }$.

Figure 8 shows the leverage structure for the predictor variables of this data set based on the robust MCD based distance as well as the classical one. We give results for the full data set and a reduced data set (without the collinearity hiding observation 24).

The results for the full data set are given in Table 4. Under penalized unweighted RQ, β is reasonably best estimated at τ=0.25 while, under penalized, WQR, it is reasonably best estimated at τ=0.5 across all penalty functions. Under penalized WQR, the model with tuning parameter λ=0 is selected indicating that unpenalized WQR exhibits the optimal model. There is an improvement in adopting unpenalized WQR compared to penalized QR at both τ-levels. However, the best parameter estimation is exhibited under unpenalized WQR at τ=0.5.

The results for the reduced data set without observation 24 are given in Table 5, leaving only one mild leverage point, observation 8. Cross validation results have consistently chosen unpenalized QR and unpenalized WQR models as the optimal models except for QR-LASSO, QR-E-NET, and WRQ-LASSO at τ=0.5, where λ≠0. The best results are exhibited under unpenalized WQR at τ=0.5 followed by unpenalized WQR at τ=0.25.

## 6. Conclusions

This paper suggested a penalized WQR procedure making use of robust weights based on the computationally intensive high breakdown MCD method rather than the well-known classical Mahalanobis distance or any other LS based weights as in [20] which are amenable to outliers. As penalty functions, the RIDGE, LASSO, and E-NET penalties were used yielding the procedures WQR-RIDGE, WQR-LASSO, and the WQR-E-NET, respectively. The efficacy of these procedures as a remedy to high leverage points and collinearity influential points were investigated via simulations and applications to well-known data sets from the literature.

Simulation studies show that generally penalized versions of robustly WQR performs better than their unweighted counterparts, with the WQR-LASSO generally performing the best although marginally so at D2 and D4 under the Normal distribution. However, there are few exceptions; at D2 and D4 under the Normal distribution, with respect to model/variable selection, WQR-LASSO and WQR-E-NET alternately dominate each other while, with respect to prediction, penalized QR performs better than penalized WQR. The occasional dominance of the WQR-E-NET over WQR-LASSO is expected (see, e.g., [11]).

Applications to well-known data sets from the literature show that, in some cases, applying the MCD based robust weight is adequate, i.e., WQR (tuning parameter λ=0) performs better than penalized WQR. The best performance is mostly at a quantile level τ=0.5 while WQR-LASSO and WQR-E-NET are too “greedy” at τ=0.25, shrinking all parameters to zero. Thus, overall, simulations and applications to the [35] and the [36] data sets show an improvement in variable selection and regularization due to the robust weighting formulation.

## Figures and Tables

**Figure 1 entropy-23-00033-f001:**
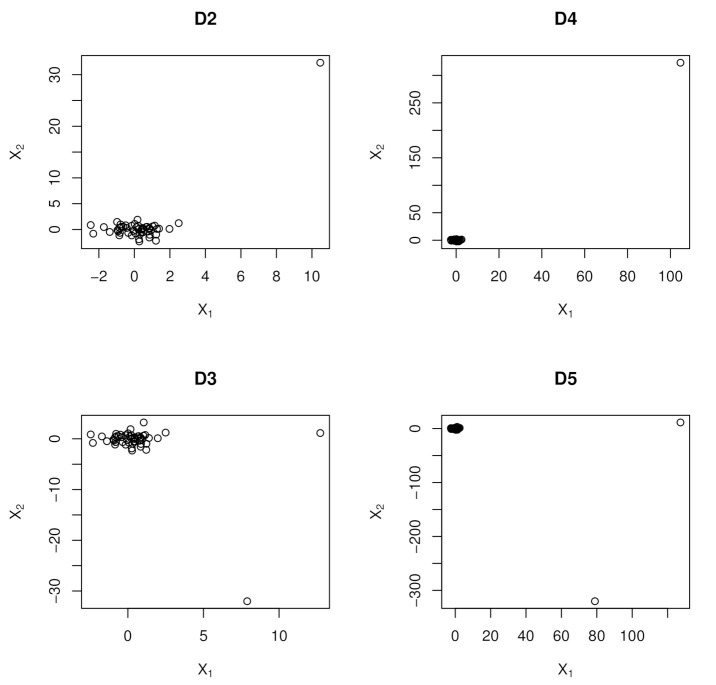
First Row Panel: Comprise Collinearity Influential Points that create collinearity in D2 and D4; Second Row Panel: Comprise Collinearity Influential Points that hide collinearity in D3 and D5.

**Figure 2 entropy-23-00033-f002:**
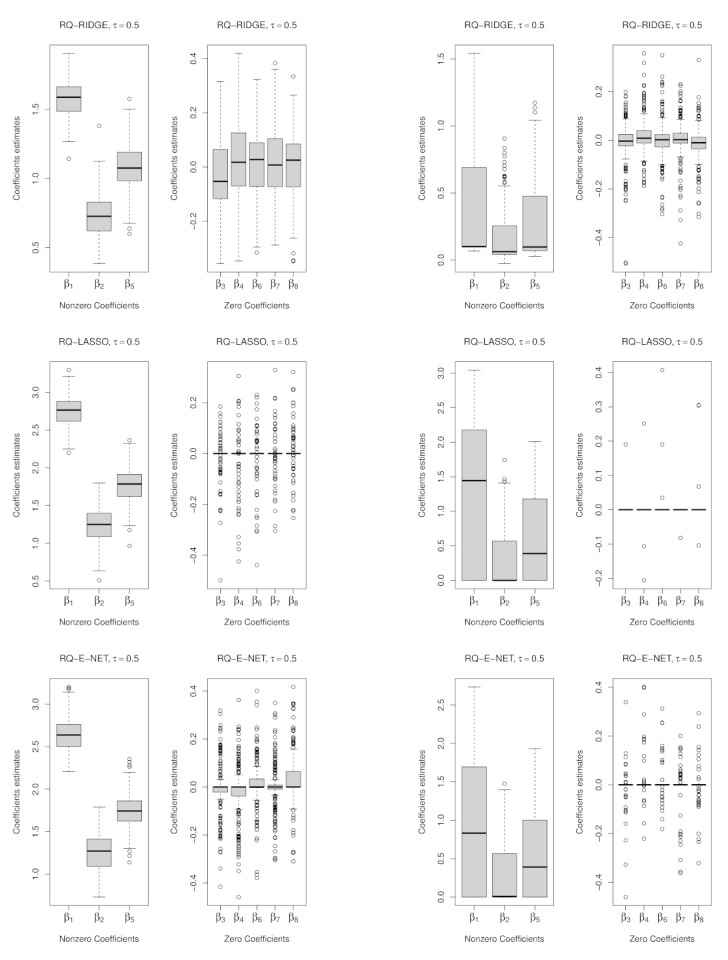
Box Plots at D1 for RQ: Left panel; under the Normal distribution with σ=1, Right panel; Under the *t*-distribution with σ=1 and d=1, τ=0.5.

**Figure 3 entropy-23-00033-f003:**
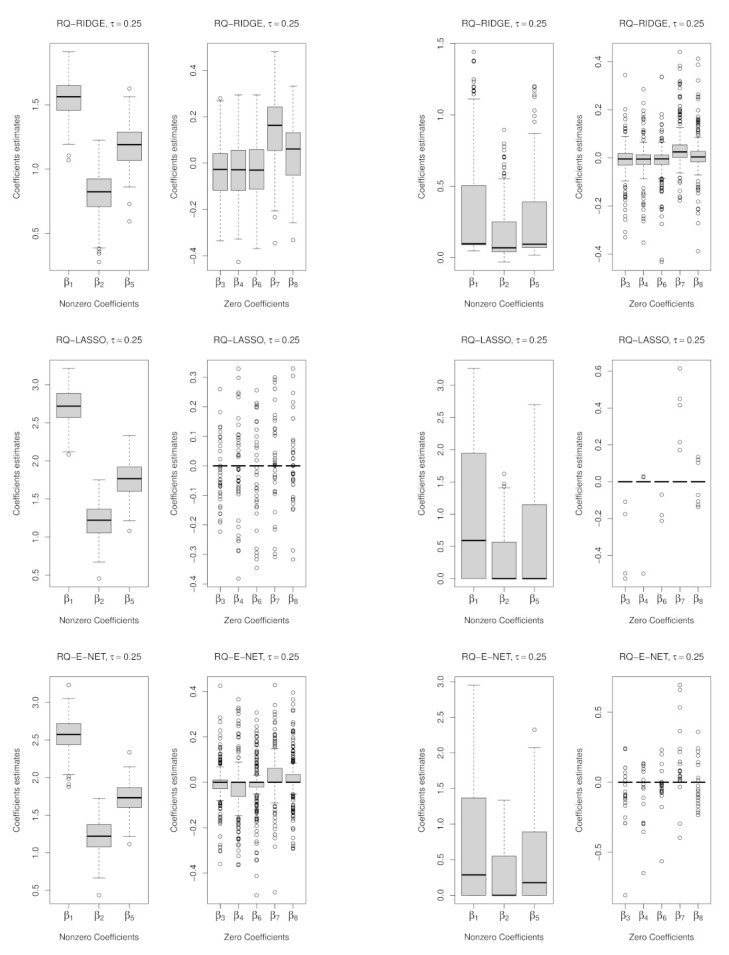
Box Plots at D1 for RQ: Left panel; under the Normal distribution with σ=1, Right panel; under the *t*-distribution with σ=1 and d=1, τ=0.25.

**Figure 4 entropy-23-00033-f004:**
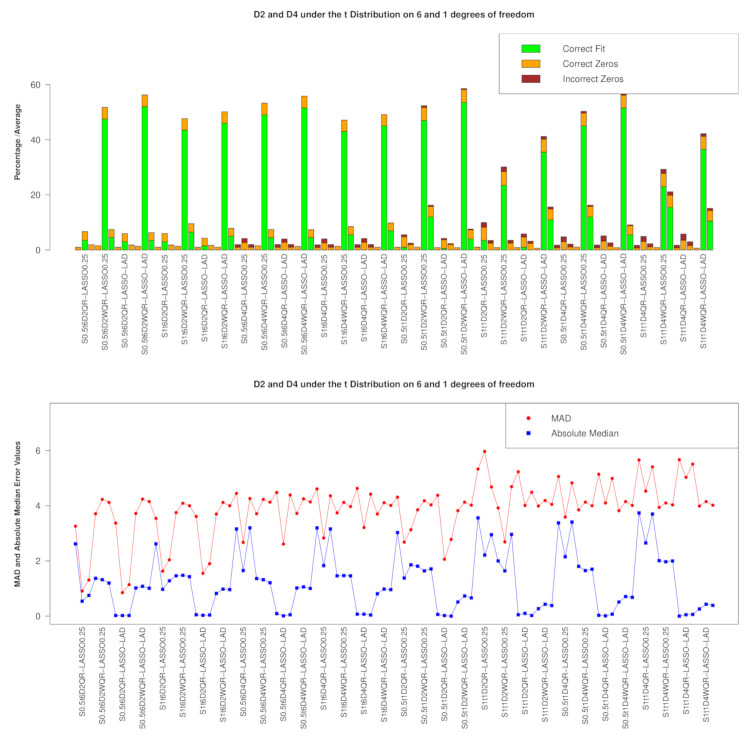
Performance at D2 and D4 under the $t_{6}$ and $t_{1}$ distributions with the RIDGE and E-NET on the LHS and RHS of LASSO, respectively; Upper panel: Model/Variable selection showing the proportion of correct models and the average of correct/incorrect βs selected; Lower panel: Prediction metrics.

**Figure 5 entropy-23-00033-f005:**
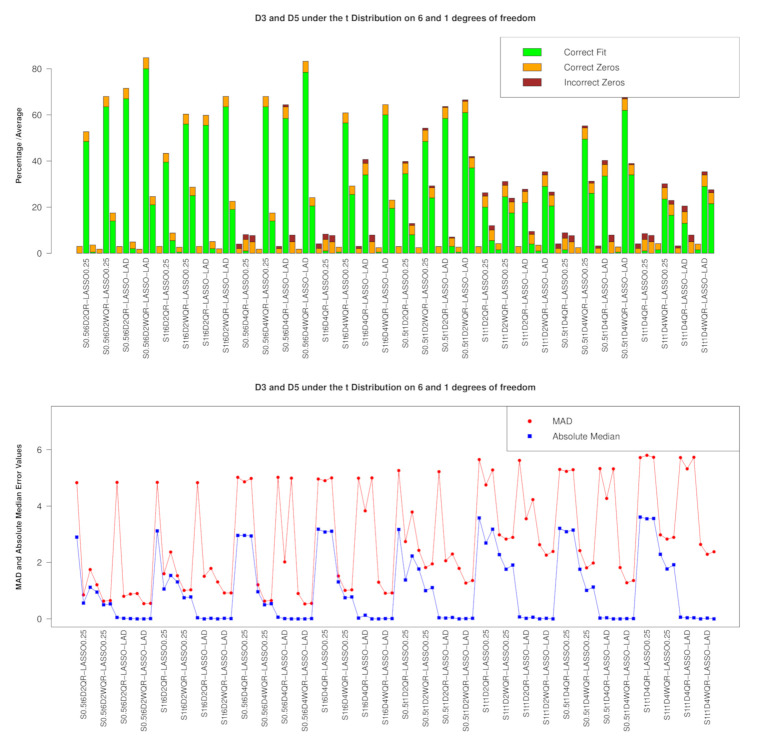
Performance at D3 and D5 under the $t_{6}$ and $t_{1}$ distributions distributions with the RIDGE and E-NET on the LHS and RHS, respectively; Upper panel: Model/Variable selection showing the proportion of correct models and the average of correct/incorrect βs selected; Lower panel: Prediction metrics.

**Figure 6 entropy-23-00033-f006:**
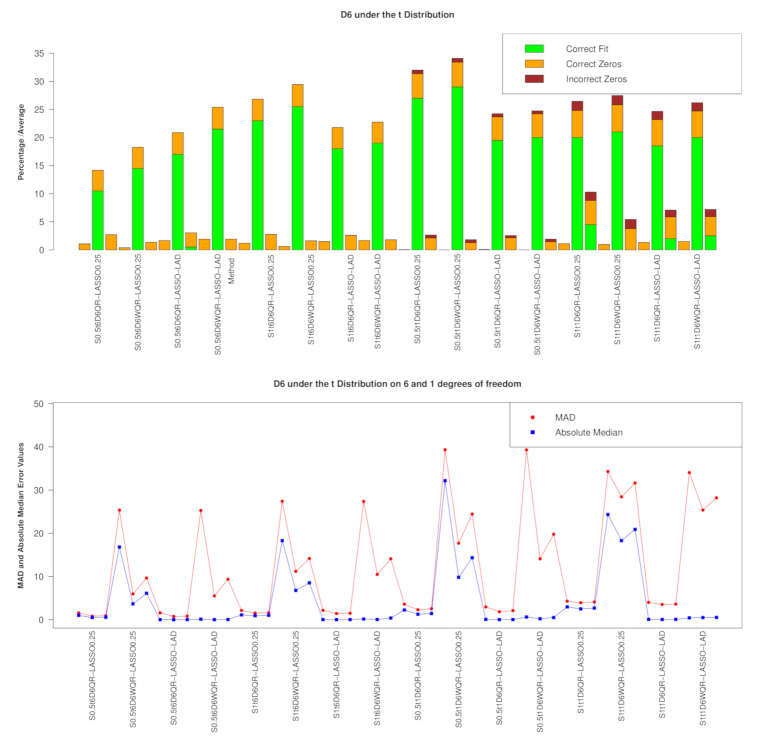
Performance at D6 under the $t_{6}$ and $t_{1}$ distributions with the RIDGE and E-NET on the LHS and RHS of LASSO, respectively; Model/Variable selection showing the proportion of correct models and the average of correct/incorrect βs selected; Lower panel: Prediction metrics.

**Figure 7 entropy-23-00033-f007:**
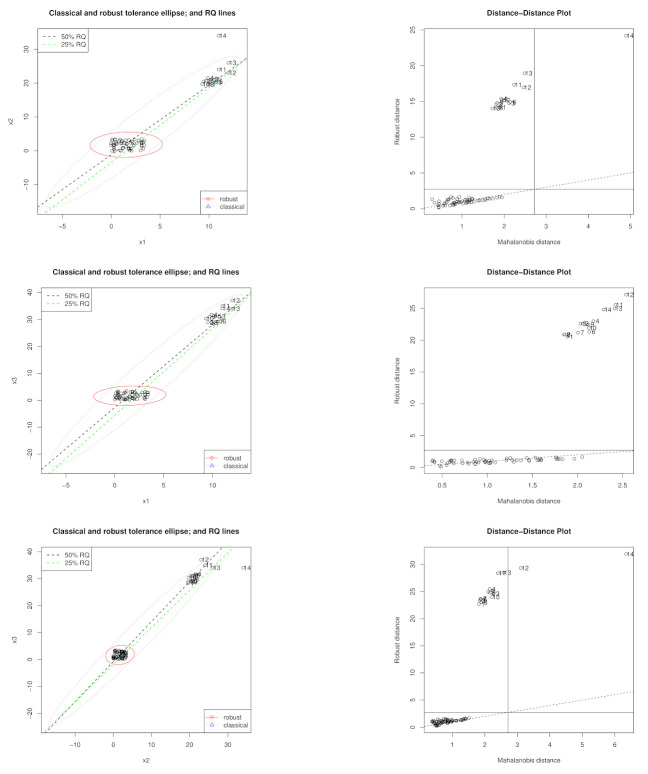
Tolerance ellipses and distance–distance plots for the [35] data set.

**Figure 8 entropy-23-00033-f008:**
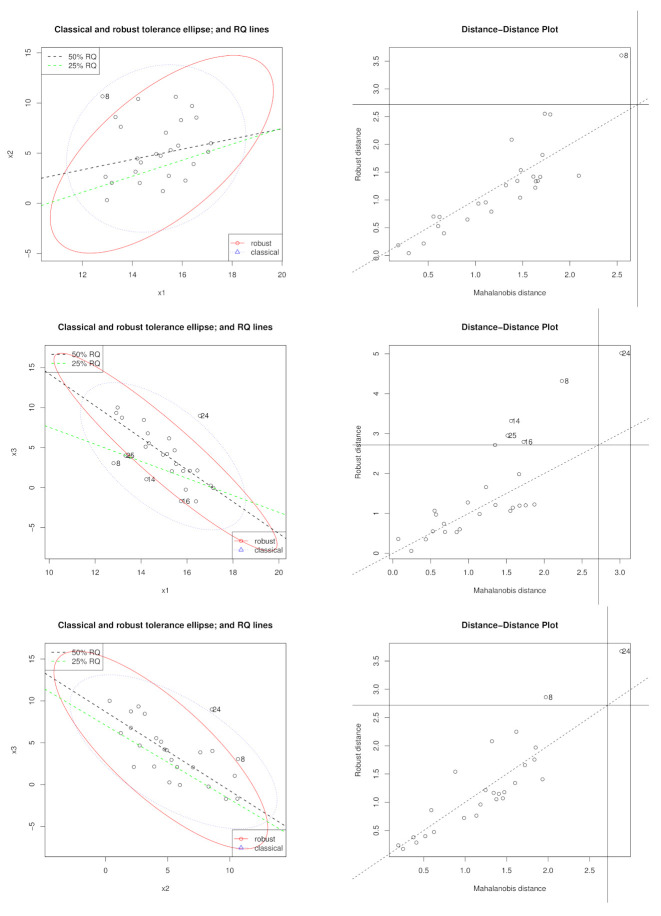
Tolerance Ellipses and Distance-Distance Plots for the [36] data set.

**Table 1 entropy-23-00033-t001:** Quantile regression at D1 (at quantile levels τ=0.5 and 0.25) for n=50.

			Correctly	No. of Zeros		Med (MAD)	
	Parameters	Method	Fitted	Correct	Incorrect	Test Error	Optimal λ
D1 under the Normal Distribution	σ=1, τ=0.25	QR-RIDGE	0	2.27	0	1.28(1.97)	0.12
		QR-LASSO	**67.5**	**4.56**	0	**0.71(1.20)**	0.04
		QR-E-NET	18.5	3.59	0	0.72(1.25)	0.04
	σ=1, τ=0.5	QR-RIDGE	1.5	2.33	0	−0.03(1.99)	0.14
		QR-LASSO	**62**	**4.49**	0	**0.00(1.15)**	0.05
		QR-E-NET	24	3.6	0	0.01(1.19)	0.04
	σ=3, τ=0.25	QR-RIDGE	9	3.07	0.03	2.70(4.32)	0.12
		QR-LASSO	**39.5**	**4.52**	0.38	**2.03(3.60)**	0.04
		QR-E-NET	30.5	4	0.2	2.18(3.69)	0.04
	σ=3, τ=0.5	QR-RIDGE	2.5	2.37	0.01	−0.04(4.06)	0.12
		QR-LASSO	**40**	**4.57**	0.32	**0.01(3.45)**	0.05
		QR-E-NET	31	3.9	0.11	0.00(3.55)	0.04
D1 under the *t* Distribution	d=1, σ=0.5,	QR-RIDGE	3.00	2.33	0.02	2.17(3.21)	0.11
	τ=0.25	QR-LASSO	**64.00**	**4.92**	0.72	**1.24(2.16)**	0.04
		QR-E-NET	36.50	4.42	0.62	1.44(2.41)	0.03
	d=1, σ=0.5,	QR-RIDGE	1.50	2.56	0.01	0.02(2.94)	0.13
	τ=0.5	QR-LASSO	**64.50**	**4.94**	0.67	**0.02(1.72)**	0.04
		QR-E-NET	32.50	4.33	0.58	−0.01(1.94)	0.03
	d=1, σ=1,	QR-RIDGE	2.50	2.37	0.03	3.05(4.30)	0.11
	τ=0.25	QR-LASSO	**30.50**	**4.95**	1.57	**2.44(3.80)**	0.04
		QR-E-NET	26.00	4.78	1.49	2.62(4.04)	0.03
	d=1, σ=1,	QR-RIDGE	3.50	2.49	0.02	0.02(4.15)	0.12
	τ=0.5	QR-LASSO	**33.50**	**4.95**	1.38	**0.02(3.34)**	0.04
		QR-E-NET	25.50	4.66	1.27	0.02(3.58)	0.03

**Table 2 entropy-23-00033-t002:** Results for the full [35] data set.

		β	UNWEIGHTED							
			NONE-BIASED		RIDGE		LASSO		E-NET	
			$\hat{\beta }$	Bias	$\hat{\beta }$	Bias	$\hat{\beta }$	Bias	$\hat{\beta }$	Bias
λ			0.00		0.00		0.00		0.00	
RQ τ=0.5	intercept	0.00	2.27	−2.27	2.39	−2.39	2.39	−2.39	2.39	−2.39
	$X_{1}$	2.00	1.39	0.61	1.45	0.55	1.45	0.55	1.45	0.55
	$X_{2}$	2.00	1.87	0.13	1.79	0.21	1.79	0.21	1.79	0.21
	$X_{3}$	0.00	−0.78	0.78	−0.74	0.74	−0.74	0.74	−0.74	0.74
λ			0.00		0.00		0.00		0.00	
RQ τ=0.25	intercept	−1.00	1.09	−2.09	1.32	−2.32	1.32	−2.32	1.32	−2.32
	$X_{1}$	2.00	1.59	0.41	1.48	0.52	1.48	0.52	1.48	0.52
	$X_{2}$	2.00	1.94	0.06	1.80	0.20	1.80	0.20	1.80	0.20
	$X_{3}$	0.00	−0.88	0.88	−0.76	0.76	−0.76	0.76	−0.76	0.76
		β			**WEIGHTED**					
			**NONE-BIASED**		**RIDGE**		**LASSO**		**E-NET**	
			$\hat{\beta }$	Bias	$\hat{\beta }$	Bias	$\hat{\beta }$	Bias	$\hat{\beta }$	Bias
λ			0.00		0.00		0.06		0.00	
RQ τ=0.5	intercept	0.00	2.27	−2.27	0.11	−0.11	0.00	0.00	0.11	−0.11
	$X_{1}$	2.00	1.39	0.61	1.93	0.07	1.93	0.07	1.93	0.07
	$X_{2}$	2.00	1.87	0.13	2.01	−0.01	1.97	0.03	2.01	−0.01
	$X_{3}$	0.00	−0.78	0.78	−0.09	0.09	0.00	0.00	−0.09	0.09
λ			0.00		0.50		0.50		0.50	
RQ τ=0.25	intercept	−1.00	1.09	−2.09	0.18	−1.18	0.29	−1.29	0.29	−1.29
	$X_{1}$	2.00	1.59	0.41	0.35	1.65	0.00	2.00	0.00	2.00
	$X_{2}$	2.00	1.94	0.06	0.39	1.61	0.00	2.00	0.00	2.00
	$X_{3}$	0.00	−0.88	0.88	0.38	−0.38	0.00	0.00	0.00	0.00

**Table 3 entropy-23-00033-t003:** Results for [35]; without observations 1–14.

		β	UNWEIGHTED							
			NONE-BIASED		RIDGE		LASSO		E-NET	
			$\hat{\beta }$	Bias	$\hat{\beta }$	Bias	$\hat{\beta }$	Bias	$\hat{\beta }$	Bias
λ			0.00		0.00		0.00		0.00	
RQ τ=0.5	*intercept*	0.00	0.74	−0.74	0.50	−0.50	0.50	−0.50	0.50	−0.50
	$X_{1}$	2.00	1.86	0.14	1.84	0.16	1.84	0.16	1.84	0.16
	$X_{2}$	2.00	1.93	0.07	1.87	0.13	1.87	0.13	1.87	0.13
	$X_{3}$	0.00	−0.07	0.07	−0.03	0.03	−0.03	0.03	−0.03	0.03
λ			0.00		0.00		0.00		0.00	
RQ τ=0.25	*intercept*	−1.00	0.69	−1.69	−0.09	−0.91	−0.09	0.09	−0.09	−0.91
	$X_{1}$	2.00	1.67	0.33	1.73	0.27	1.73	0.27	1.73	0.27
	$X_{2}$	2.00	1.90	0.10	1.95	0.05	1.95	0.05	1.95	0.05
	$X_{3}$	0.00	−0.08	0.08	−0.10	0.10	−0.10	0.10	−0.10	0.10
		β			**WEIGHTED**					
			**NONE-BIASED**		**RIDGE**		**LASSO**		**E-NET**	
			$\hat{\beta }$	Bias	$\hat{\beta }$	Bias	$\hat{\beta }$	Bias	$\hat{\beta }$	Bias
λ			0.00		0.00		0.00		0.00	
RQ τ=0.5	*intercept*	0.00	0.74	−0.74	0.27	−0.27	0.27	−0.27	0.27	−0.27
	$X_{1}$	2.00	1.86	0.14	1.86	0.14	1.86	0.14	1.86	0.14
	$X_{2}$	2.00	1.93	0.07	1.96	0.04	1.96	0.04	1.96	0.04
	$X_{3}$	0.00	−0.07	0.07	−0.06	0.06	−0.06	0.06	−0.06	0.06
λ			0.00		0.50		0.33		0.50	
RQ τ=0.25	*intercept*	−1.00	0.69	−1.69	1.74	−2.74	2.22	−3.22	2.20	−3.20
	$X_{1}$	2.00	1.67	0.33	0.30	1.70	0.00	2.00	0.00	2.00
	$X_{2}$	2.00	1.90	0.10	0.48	1.52	0.00	2.00	0.10	1.90
	$X_{3}$	0.00	−0.08	0.08	0.22	−0.22	0.00	0.00	0.00	0.00

**Table 4 entropy-23-00033-t004:** Results for the full [36] data set.

		β	UNWEIGHTED							
			NONE-BIASED		RIDGE		LASSO		E-NET	
			$\hat{\beta }$	Bias	$\hat{\beta }$	Bias	$\hat{\beta }$	Bias	$\hat{\beta }$	Bias
λ			0.00		0.11		0.06		0.11	
RQ τ=0.5	*intercept*	0.00	25.09	−25.09	24.34	−24.34	27.63	−27.63	23.63	−23.63
	$X_{1}$	3.00	1.55	1.45	0.86	2.14	1.28	1.72	1.06	1.94
	$X_{2}$	−2.00	−2.30	0.30	−0.86	−1.14	−2.12	0.12	−1.21	−0.79
	$X_{3}$	0.00	−0.66	0.66	0.17	−0.17	−0.49	0.49	0.00	0.00
λ			0.00		0.00		0.06		0.06	
RQ τ=0.25	*intercept*	−1.00	23.53	−24.53	25.26	−26.26	30.32	−31.32	33.13	−34.13
	$X_{1}$	3.00	1.19	1.81	1.09	1.91	0.56	2.44	0.30	2.70
	$X_{2}$	−2.00	−1.96	−0.04	−1.98	−0.02	−1.70	−0.30	−1.53	−0.47
	$X_{3}$	0.00	−0.15	0.15	−0.16	0.16	0.00	0.00	0.02	−0.02
		β			**WEIGHTED**					
			**NONE-BIASED**		**WQR-RIDGE**		**WQR-LASSO**		**WQR-E-NET**	
			$\hat{\beta }$	Bias	$\hat{\beta }$	Bias	$\hat{\beta }$	Bias	$\hat{\beta }$	Bias
RQ τ=0.5	*intercept*	0.00	25.09	−25.09	0.36	−0.36	0.36	−0.36	0.36	−0.36
	$X_{1}$	3.00	1.55	1.45	2.94	0.06	2.94	0.06	2.94	0.06
	$X_{2}$	−2.00	−2.30	0.30	−2.08	0.08	−2.08	0.08	−2.08	0.08
	$X_{3}$	0.00	−0.66	0.66	0.01	−0.01	0.01	−0.01	0.01	−0.01
λ			0.00		0.00		0.00		0.00	
RQ τ=0.25	*intercept*	−1.00	23.53	−24.53	−0.08	−0.92	−0.08	−0.92	7.62	−8.62
	$X_{1}$	3.00	1.19	1.81	2.95	0.05	2.95	0.05	2.95	0.05
	$X_{2}$	−2.00	−1.96	−0.04	−2.47	0.47	−2.47	0.47	−2.47	0.47
	$X_{3}$	0.00	−0.15	0.15	−0.03	0.03	−0.03	0.03	−0.03	0.03

**Table 5 entropy-23-00033-t005:** Results for [36] data set without observation 24.

		β	UNWEIGHTED							
			NONE-BIASED		RIDGE		LASSO		E-NET	
			$\hat{\beta }$	Bias	$\hat{\beta }$	Bias	$\hat{\beta }$	Bias	$\hat{\beta }$	Bias
λ			0.00		0.00		0.22		0.08	
RQ τ=0.5	*intercept*	0.00	−59.31	59.31	−56.47	56.47	40.67	−40.67	8.77	−8.77
	$X_{1}$	3.00	5.78	−2.78	5.65	−2.65	0.00	3.00	2.09	0.91
	$X_{2}$	−2.00	−0.22	−1.78	−0.32	−1.68	−1.18	−0.82	−1.37	−0.63
	$X_{3}$	0.00	2.13	−2.13	2.05	−2.05	0.00	0.00	0.30	−0.30
λ			0.00		0.00		0.00		0.00	
RQ τ=0.25	*intercept*	−1.00	−56.16	55.16	−59.60	58.60	−59.61	58.61	−59.61	58.61
	$X_{1}$	3.00	5.67	−2.67	5.80	−2.80	5.80	−2.80	5.80	−2.80
	$X_{2}$	−2.00	−0.61	−1.39	−0.48	−1.52	−0.48	−1.52	−0.48	−1.52
	$X_{3}$	0.00	1.96	−1.96	2.13	−2.13	2.13	−2.13	2.13	−2.13
		β			**WEIGHTED**					
			**NONE-BIASED**		**WQR-RIDGE**		**WQR-LASSO**		**WQR-E-NET**	
			$\hat{\beta }$	Bias	$\hat{\beta }$	Bias	$\hat{\beta }$	Bias	$\hat{\beta }$	Bias
λ			0.00		0.00		0.06		0.00	
RQ τ=0.5	*intercept*	0.00	−59.31	59.31	0.12	−0.12	−0.24	0.24	0.12	−0.12
	$X_{1}$	3.00	5.78	−2.78	2.88	0.12	2.77	0.23	2.88	0.12
	$X_{2}$	−2.00	−0.22	−1.78	−1.88	−0.12	−1.44	−0.56	−1.88	−0.12
	$X_{3}$	0.00	2.13	−2.13	0.07	−0.07	0.20	−0.20	0.07	−0.07
λ			0.00		0.00		0.00		0.00	
RQ τ=0.25	*intercept*	−1.00	−56.16	55.16	−0.37	−0.63	−0.37	−0.63	−0.37	−0.63
	$X_{1}$	3.00	5.67	−2.67	3.02	−0.02	3.02	−0.02	3.02	−0.02
	$X_{2}$	−2.00	−0.61	−1.39	−2.59	0.59	−2.59	0.59	−2.59	0.59
	$X_{3}$	0.00	1.96	−1.96	−0.12	0.12	−0.12	0.12	−0.12	0.12

## Data Availability

Not applicable.

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
