# Peer review of "Variable Selection and Regularization in Quantile Regression via Minimum Covariance Determinant Based Weights"

_entropy, 2020, doi:10.3390/e23010033_

Round 1

Reviewer 1 Report

The MS aims at robustification of quantile regression using MCD-based weights which should suppress the influence of leverage points (outliers in direction of x-axis). From this perspective, it is not a breakthrough idea, but as it is quite direct, the results are as expected. Though I have two major doubts and some minor remarks:

(1) In lines 126-129 some asymptotic results are listed but without proving a formal proof. This needs to be done, or clearly set (including reference!) why such properties follow immediately.
(2) The simulation study and examples (Sections 4 and 5) with numbers of plots and tables sound weird for me. In robustness literature it is usual that the simulation settings have some two or three parameters which are continuously changed and behaviour of methods is compared. Your simulation needs to be redone accordingly by condensing the toy simulation settings D2-D5. Also in examples I would condense the resulting tables only to those which show main properties of the methods.

Minor comments:

l.18: are an important considerations of -> are popularly considered in (?)
l.41: are, the -> are the
l.43: "The author [14] developed upon [19]’s work" sounds strange.
l.47: the the -> the
l.61: mahalanobis -> Mahalanobis (throughout the paper)
l.82: equation -> Equation; "3" should be "2".
l.88: equation 5 -> Equation 3
l.91: terms respectively.In -> terms, respectively. In
l.96: Lower indices in x_i's in the hat matrix should be not in bold.
l.102: weight [30] -> weight, [30]
l.106: section -> Equation
l.120: to be true as stated in [13] as follows -> are fulfilled, see [13] for details
l.123: What does mean O(1)?
l.124: Replace the initial symbol by "There exists"
l.134: X'X=nI - do you mean the covariance matrix?
l.135: euclidian - Euclidean (throughout the paper)
l.155: Figure 2 -> Figure 1
l.162: Why 100-fold cross validation when the usual choice is 5-fold or 10-fold CV?
Delete Section 7 and Appendices A and B which remained from the template.

Author Response

Please find response to reviewer 1 attached.

Reviewer 2 Report

see attachment

Author Response

Please find responses to reviewer 2 attached.

Round 2

Reviewer 1 Report

The MS was substantially improved wrt. previous version. Some remaining small observations follow:

line 137: exist -> exists
line 165: Euclidian -> Euclidean
line 190: lambda should be a Greek letter
It is strange to see the distorted tables, but I guess that they will be either corrected or removed.

Reviewer 2 Report

I can see that you have worked hard on the edits to this paper. A noticed some minor issues when reviewing the text. 

1) There is a line on page 7 between line 164 and 165 of X ~ N(0,D2

The bracket is not closed and the text is strange in that area. 

2) Most of the Tables runoff the page. I suspect that could be solved by actually deleting the text that needs to be deleted instead of crossing it out in red.